# Does Transcranial Direct Current Stimulation Affect Potential P300-Related Events in Vascular Dementia? Considerations from a Pilot Study

**DOI:** 10.3390/biomedicines12061290

**Published:** 2024-06-11

**Authors:** Antonio Gangemi, Rosa Angela Fabio, Rossella Suriano, Rosaria De Luca, Angela Marra, Mariangela Tomo, Angelo Quartarone, Rocco Salvatore Calabrò

**Affiliations:** 1IRCCS Centro Neurolesi “Bonino-Pulejo”, S.S. 113, Cda Casazza, 98124 Messina, Italy; antonio.gangemi@irccsme.it (A.G.); rosaria.deluca@irccsme.it (R.D.L.); angela.marra@irccsme.it (A.M.); angelo.quartarone@irccsme.it (A.Q.); roccos.calabro@irccsme.it (R.S.C.); 2Department of Cognitive, Psychological and Pedagogical Sciences and Cultural Studies, University of Messina, 98100 Messina, Italy; rafabio@unime.it; 3Grande Ospedale Metropolitano “Bianchi-Melacrino-Morelli”, 89124 Reggio Calabria, Italy; mariangelatomo4@gmail.com

**Keywords:** vascular dementia, transcranial direct current stimulation (tDCS), electroencephalogram (EEG), P300 event-related potential

## Abstract

Vascular dementia, the second most common type of dementia, currently lacks a definitive cure. In the pursuit of therapies aimed at slowing its progression and alleviating symptoms, transcranial direct current stimulation (tDCS) emerges as a promising approach, characterized by its non-invasive nature and the ability to promote brain plasticity. In this study, the primary objective was to investigate the effects of a two-week cycle of tDCS on the dorsolateral prefrontal cortex (DLPFC) and neurophysiological functioning in thirty patients diagnosed with vascular dementia. Each participant was assigned to one of two groups: the experimental group, which received anodal tDCS to stimulate DPCFL, and the control group, which received sham tDCS. Neurophysiological functions were assessed before and after tDCS using P300 event-related potentials (ERPs), while neuropsychological function was evaluated through a Mini-Mental State Examination (MMSE). The results showed a reduction in P300 latency, indicating a faster cognitive process; an increase in P300 amplitude, suggesting a stronger neural response to cognitive stimuli; and a significant improvement in MMSE scores compared to the control group, indicating an overall enhancement in cognitive functions. These findings suggest that tDCS could represent a promising therapeutic option for improving both neurophysiological and cognitive aspects in patients with vascular dementia.

## 1. Introduction

Vascular dementia is a neurodegenerative condition resulting from brain damage caused by disruptions in blood flow to the brain, leading to significant impacts on cognitive and psychomotor functions [1,2,3]. It represents the second most common type of dementia after Alzheimer’s disease (AD), affecting approximately 46.8 million individuals worldwide and contributing to at least 20–25% of total dementia cases [4,5]. Its incidence doubles every 5–10 years from the age of sixty-five [6]. Aging is the primary risk factor for the development of vascular dementia, followed by common vascular risk factors such as hypertension, diabetes mellitus, smoking, and atrial fibrillation [7,8]. These factors contribute to the progressive deterioration of the cerebral vascular system, involving arteries of assorted sizes, leading to neurodegeneration and cognitive decline [9,10,11]. Vascular dementia can result from damage caused by acute ischemic events, such as strokes, or chronic brain injuries due to prolonged reduced blood perfusion [12,13]. Its clinical manifestations can vary depending on the location, size, and severity of vascular brain lesions [14]. Vascular dementia encompasses a range of cognitive disorders, ranging from mild cognitive deficits to severe dementia manifestations [15,16]. This clinical complexity significantly impacts the quality of life for affected individuals and their families [17,18,19].

Pharmacological treatment options for vascular dementia include cholinesterase inhibitors, glutamate receptor antagonists, vitamins, and antiplatelet agents [20,21,22]. However, such pharmacological therapies often prove unsatisfactory in clinical practice and are associated with significant side effects, such as excessive salivation, sweating, gastrointestinal disturbances, restlessness, and dizziness [23,24]. Most interventions primarily focus on controlling risk factors, relieving symptoms, and providing support to patients and their families [25,26,27].

As a result, research has explored the potential efficacy of non-pharmacological therapeutic approaches, such as non-invasive brain stimulation (NIBS) [28,29]. Specifically, transcranial direct current stimulation (tDCS) modulates neural activity, promoting synaptic plasticity and the consolidation of compensatory brain networks [30,31,32]. It can influence cortical activity both in the short term, by regulating resting membrane potential, and in the long term, by modulating synaptic plasticity [33,34]. It is considered a safe rehabilitative strategy, with a low incidence of side effects and characterized by painless application and cost-effectiveness [34,35]. tDCS has achieved notable success and promising results in treating degenerative brain processes, including those related to physiological aging and pathological conditions [36,37,38,39]. Furthermore, the application of tDCS has shown promise in managing mild cognitive decline and dementia [40,41,42]. Studies conducted on patients with vascular dementia, using sessions of anodal or sham tDCS with an intensity of 2 mA targeted at stimulating the left DLPFC, revealed clinically significant improvements up to two weeks after the conclusion of the sessions. Among the positive outcomes, improvements in visual recall and reaction times during tasks such as the n-back and go/no-go tests stand out. The analysis of these results suggests a differentiated impact of tDCS on specific aspects of cognitive and behavioral functions in patients with vascular dementia [43,44,45].

To assess the neural process involved in cognitive activities and to monitor changes in cognitive function, it is possible to employ event-related potentials (ERPs) recorded through electroencephalography (EEG) [46,47]. In the context of dementia, particular importance is attributed to the P300 component of ERP, associated with the cognitive processes related to attention, memory, and decision making [48,49,50]. The latency of P300 represents a crucial indicator of the speed and efficiency of cognitive processing [51,52]. The amplitude of P300 reflects the intensity of the brain’s response to significant stimuli [53,54,55]. Increased amplitude suggests a stronger response, indicating greater cognitive resource engagement. Although there is evidence of the effectiveness of ERPs in monitoring symptomatic improvements resulting from tDCS [56,57,58], there is a lack of research in this area. While some studies have investigated the neurophysiological effects of tDCS in individuals with mild cognitive decline [59,60,61], few studies have been conducted on the patients with vascular dementia.

In the present study, it is hypothesized that the application of tDCS may induce significant modifications in neurophysiological parameters among the patients with vascular dementia. Specifically, we anticipate that tDCS intervention will influence the amplitude and latency of the P300 component in response to cognitive stimuli. The P300 is commonly associated with cognitive processing, and the central hypothesis posits that tDCS can modulate these parameters, reflecting potential improvements in cognitive functions in individuals with vascular dementia. Additionally, we hypothesize that the application of tDCS will have a positive impact on the Mini-Mental State Examination (MMSE) scores in patients with vascular dementia. The MMSE serves as a widely used indicator for assessing cognitive decline and dysfunction. The hypothesis is grounded in the idea that tDCS, by acting on neural circuits involved in cognitive functions, may result in measurable improvements in cognitive performance as assessed by the MMSE. The underlying basis for the primary hypothesis is rooted in tDCS ability to modulate neuronal activity in specific brain regions involved in cognitive processes, including those associated with P300 generation. The effects of tDCS on synaptic plasticity and neuronal function could manifest in the electrophysiological parameters of the P300, providing an objective indicator of the intervention’s effects on cognitive response. The secondary hypothesis is supported by the notion that vascular dementia is characterized by structural and functional alterations in the brain, and tDCS might mitigate such dysfunctions through the modulation of neuronal activity. The expected effect on the MMSE scale assumes that improvements in cognitive function, reflected in P300 parameters, will translate into a higher overall score on the assessment of cognitive abilities.

## 2. Materials and Methods

### 2.1. Participants

Thirty patients diagnosed with vascular dementia were invited to participate in the study [62]. The participants were recruited from the IRCCS Neurolesi “Bonino-Pulejo” in Messina, located in southern Italy, from January to July 2023. Inclusion criteria were as follows: (a) a clinical diagnosis of vascular dementia, according to the established diagnostic criteria such as the Diagnostic and Statistical Manual of Mental Disorders (DSM) or the International Classification of Diseases (ICD); (b) age range 50–80; and (c) evidence of vascular lesions in the brain, confirmed by computed tomography (CT) or magnetic resonance imaging (MRI). Exclusion criteria were as follows: (i) patients with cranial metallic implants; (ii) previous neurosurgical interventions; (iii) patients with psychiatric, neurodegenerative, epilepsy, or inflammatory brain diseases; (iv) patients with significant sensory impairments such as severe vision and hearing impairment. The meticulous selection of participants contributed to ensuring the reliability and validity of the data collected during the study. The present investigation was conducted in accordance with the principles set forth in the Declaration of Helsinki and received approval from the Ethics Committee of the IRCCS Neurolesi Bonino Pulejo in Messina (protocol approval number IRCCS-ME CEL/U21/22 16 December 2022). Before entering the study, each participant was carefully informed about the purpose of the research and the data collection procedures. This information was presented clearly and comprehensibly to enable each participant to make an informed decision regarding their participation. After receiving such explanations, each participant provided their written informed consent. Prior to the start of the intervention, participants were fully instructed on the stimulation procedure employed in the study. This instruction included details on the functioning of tDCS and its related practical procedures. Additionally, a clear explanation of the possible side effects associated with the intervention was provided. Each participant was informed of the right to withdraw from the study at any time without suffering any negative consequences. All the participants completed the study and none of them reported any side effects. The final sample consisted of 15 + 15 patients (mean age 71.2 ± 5.6, with an age range from 64 to 78 years).

### 2.2. Procedures

Before the intervention, the participants underwent an initial assessment of their neurophysiological state using the Evoked Potential P300 (ERP). Additionally, neuropsychological functions were evaluated through the MMSE. Subsequently, each participant was assigned to one of two groups using a quasi-randomized approach (i.e., they were recruited in order of enrolment). The experimental group received anodal transcranial direct current stimulation (tDCS) to stimulate the dorsolateral prefrontal cortex (DPCFL), whereas the control group received sham tDCS. The participants received tDCS sessions lasting 20 min per day for two consecutive weeks. During and after each tDCS session, the participants were carefully monitored for any side effects or discomfort, and any adverse symptoms were promptly recorded and managed. After the intervention, the participants were reassessed with the MMSE and P300 evaluation to detect any changes in their neurophysiological and neuropsychological state compared to the initial assessment. The assessors of the training were blinded to the TDCs treatment.

In this study, the tDCS BrainSTIM device, produced by EMS S.r.l. in Bologna, Italy, was utilized. Stimulation was carried out using a pair of sponge electrodes, each with a diameter of 25 mm, previously soaked in saline solution. A constant current was delivered by a battery-powered stimulator. To activate the left DLPFC, the anode was positioned on the F7 region, while the cathodal reference electrode was placed above the right supraorbital area. The precise localization of the electrodes followed the EEG 10–20 system. The stimulation intensity was set at 2 mA (with a current density of 2.5 mA/cm^2^) for a duration of 20 min. This stimulation protocol was administered five times a week for two consecutive weeks. In the control group, the sham tDCS consisted in delivering a short period of active stimulation at the beginning of the stimulation session (e.g., 10 s at 0.1 mA, 120 s at 1 mA) followed by no stimulation for a total duration equal to the duration of the active stimulation.

### 2.3. Outcome Measures

#### 2.3.1. EEG

The EEG recordings were acquired at a sampling frequency of 500 Hz using a bandpass filter set between 0.1 Hz and 70 Hz. The SCAN software (version 4.3, Neuroscan, Compumedics, El Paso, TX, USA) along with the NuAMP amplifiers was employed for recording. Eighteen scalp electrodes (Ag/AgCl) were positioned according to the standard 10/20 system described by Jasper [63]. We used a monopolar montage; therefore, the active electrodes FZ-CZ-PZ were referenced to a common electrode placed on the mastoids in position A1. Electrode impedances were kept at or below 10 kΩ. The EEG data intended for the ERP analysis underwent offline processing, involving the application of a low-pass filter at 20 Hz, baseline correction, and the segmentation of waveforms into epochs centered on stimulus presentation. Trials deviating in amplitude beyond ±100 μV were excluded. A minimum of 20 trials for each stimulus was deemed necessary for the inclusion of the individual ERP waveforms. Epochs ranging from 200 ms to 1000 ms were generated offline, centered on low and high tones along with novel sounds. P300 evoked potentials were automatically detected within specific time intervals (70–110 ms, 210–270 ms, and 270–370 ms) from midline positions (Fz, Cz, and Pz), where these peaks exhibit maximum activity. Frequent stimuli immediately preceding each infrequent stimulus were selected for averaging, ensuring comparable signal–noise ratios.

#### 2.3.2. Auditory Oddball Paradigm

The auditory oddball task involves exposure to auditory stimuli without the need to provide direct responses. Each participant was seated in a partially illuminated and soundproof room, facing a monitor positioned approximately 70 cm away. Two speakers were placed next to the monitor for audio playback. Stimuli were presented using a specialized software provided by Neurobehavioral Systems Inc., Berkeley CA, USA. The auditory paradigm consisted of two categories of sounds: frequent pure sinusoidal tones and infrequent tones. Specifically, 20% of the stimuli consisted of infrequent tones (2 kHz), while 80% comprised frequent tones (1.5 kHz), both with a duration of 200 ms, rise and fall time of 5 ms, and a sound pressure level of 70 dB, SPL. The duration of each tone or noise was 200 milliseconds, with an asynchrony interval of 700 ms between the stimuli. The presentation was divided into two blocks, each containing 700 stimuli (560 frequent and 140 infrequent). The participants were actively engaged in attending to the rare stimulus. Overall, the time required to complete the task was approximately 20 min.

#### 2.3.3. Mini-Mental State Examination

The MMSE, developed by Folstein et al. [64], has been utilized to assess cognitive impairment in patients. This test involves a series of questions and tasks designed to evaluate various cognitive aspects, including temporal and spatial orientation, short-term memory, attention, calculation abilities, language (naming and comprehension), and visuospatial skills. The maximum score attainable is 30 points, with higher scores indicating better cognitive functioning. The test administration is structured, with standardized instructions for each section. The activities include the recall of a set of words, serial subtraction by sevens, naming familiar objects, and drawing specific geometric figures. Analyzing performance in these activities is valuable for identifying potential cognitive impairments, monitoring changes over time, and contributing to the diagnosis of conditions such as dementia.

### 2.4. Statistical Analysis

All the statistical analyses were conducted using the JASP 0.18.3 software (Jeffreys’s Amazing Statistics Program is a free multi-platform open-source statistics package (https://jasp-stats.org)). Descriptive statistics of the parameters of interest were calculated and examined for both experimental groups and the control group. A MANOVA model for repeated measures was applied with a between subject factor (group: experimental and control) and phases (T0—pre-intervention baseline, T1—post-test) for each outcome measures: P300 latency, P300 amplitude and MMSE scores. A Bonferroni correction was applied for multiple comparisons. The alpha level was set to *p* < 0.05 for all statistical tests. In the case of significant effects, the effect size of the test was reported. The effect sizes were computed and categorized according to eta squared η^2^.

## 3. Results

The MANOVA model for repeated measures was applied to assess the effects of tDCS on P300 latency and amplitude, and on MMSE scores. The analysis included a between-subject factor (group: experimental and control), a within-subject factor (phases: T0—pre-intervention baseline, T1—post-test), and a group by phases interaction.

Table 1 displays the means and standard deviations of P300 latency and amplitude in experimental and control patients, along with the *p*-values of independent sample t-tests.

The results revealed significant statistical effects of the tDCS intervention on cognitive processing speed (P300 latency) and P300 amplitude, as well as MMSE scores in the post-test phase (T1). More specifically, regarding P300 latency, a significant effect of the phase factor emerged (F(1, 29) = 31.23, *p* < 0.001, η^2^ = 0.09), indicating a difference between pre- and post-test results as highlighted in (Figure 1). The group by phase interaction also showed significant effects (F(1, 29) = 11.34, *p* < 0.01, η^2^ = 0.11). Considering P300 amplitude, a significant effect of the phase factor emerged (F(1, 29) = 17.12, *p* < 0.001, η^2^ = 0.09), indicating a statistical differences difference between pre- and post-test results (Table 1 and Figure 1 and Figure 2).

The group by phase interaction also showed significant effects (F(1, 29) = 9.88, *p* < 0.01, η^2^ = 0.11). Lastly, regarding the MMSE, a significant effect of the phase factor emerged (F(1, 19) = 13.22, *p* < 0.001, η^2^ = 0.19), indicating a statistically significant difference between pre and post-test results. The group by phase interaction also showed significant effects (F(1, 19) = 11.57, *p* < 0.001, η^2^ = 0.11), indicating that the experimental group increased MMSE scores (Figure 3).

These results collectively underscore the positive impact of tDCS on neurophysiological parameters and cognitive performance, supporting the hypothesis that tDCS can serve as a promising intervention for individuals grappling with vascular dementia. The observed changes in P300 latency, amplitude, and MMSE scores collectively suggest that tDCS may hold potential as a therapeutic avenue for addressing cognitive deficits associated with vascular dementia.

## 4. Discussion

The present study aimed to explore the effects of transcranial direct current stimulation (tDCS) on vascular dementia, focusing on potential correlates with P300 event-related potentials obtained through EEG. This study contributes to the growing evidence supporting the use of tDCS in the management of vascular dementia. The results suggest that tDCS could represent a promising therapeutic option for addressing cognitive deficits associated with vascular dementia. Its non-invasive nature and positive outcomes in terms of neurophysiological and cognitive improvements offer potential added value to currently available therapeutic options [32,65].

The findings of this study support the hypothesis that transcranial direct current stimulation (tDCS) has significant positive effects on both neurophysiological and cognitive aspects in patients with vascular dementia [44,66]. Specifically, the main hypotheses related to the P300 event-related potentials and MMSE scores were confirmed. The significant reduction in P300 latency, increased P300 amplitude, and improved MMSE scores collectively suggest that tDCS holds promise as a therapeutic intervention for individuals with vascular dementia. More specifically, with reference to the effect on P300 latency and amplitude, the observed reduction in P300 latency indicates an acceleration in cognitive processing speed. This finding aligns with previous research showing that tDCS can positively influence neural processing time [57,67]. The accelerated cognitive processing may contribute to more efficient attention, memory, and decision-making processes associated with the P300 component [68,69]. The significant increase in P300 amplitude reflects heightened neural reactivity to cognitive stimuli. This result is consistent with the idea that tDCS can modulate synaptic plasticity and enhance neural responsiveness [32,65]. The greater engagement of cognitive resources, as indicated by the increased amplitude, suggests a potential improvement in cognitive functions associated with the P300 component. With reference to the effect on MMSE scores, the improvement in MMSE scores provides additional support for the positive impact of tDCS on overall cognitive function. The MMSE is a widely used tool for assessing cognitive decline and dysfunction, and the observed increase in scores suggests a global enhancement in cognitive abilities after the tDCS intervention. The cognitive improvements measured by the MMSE are consistent with the neurophysiological changes observed in the P300 parameters. This convergence of results strengthens the argument that tDCS may have a comprehensive and beneficial impact on cognitive functions in individuals with vascular dementia [43,45]. The strength of the study is that the integrated approach combining tDCS with neurophysiological and neuropsychological assessments contributes to a comprehensive understanding of the effects of tDCS on vascular dementia. This multidimensional evaluation enhances the robustness of the findings. The exclusive use of tDCS without additional treatments minimizes confounding variables, allowing for a more direct assessment of the effects of tDCS on neurophysiological and cognitive outcomes.

The main limitation of the study is the relatively small sample size preventing us from generalizing the results. Future studies with larger and more diverse samples are warranted to ensure the external validity of the findings. The absence of adequate randomization raises questions about the validity of the study. Future research should incorporate well-designed randomized control groups to better attribute observed effects to the tDCS intervention. Long-term follow-up assessments would provide insights into the persistence of tDCS effects over time and inform the durability of cognitive improvements. Exploring alternative tDCS protocols, such as varying stimulation parameters or combining tDCS with other therapeutic approaches, may help optimize the therapeutic effects and further advance clinical applicability.

With reference to the clinical implications, the study suggests that tDCS could represent a promising non-invasive therapeutic option for managing cognitive deficits in vascular dementia. Its positive outcomes and non-invasive nature make it an attractive adjunct to existing therapeutic strategies. The potential clinical benefits of tDCS, as demonstrated by the improvements in the P300 parameters and MMSE scores, highlight its relevance in the broader context of dementia management.

## 5. Conclusions

In conclusion, this study contributes valuable evidence to the growing body of research supporting the efficacy of tDCS in the management of vascular dementia. The positive effects observed in both neurophysiological and cognitive domains underscore the potential of tDCS as a promising intervention for individuals grappling with cognitive deficits associated with vascular dementia. However, further research with larger samples, robust control groups, and long-term follow-up is essential to confirm and extend these findings, ultimately informing the development of optimized and evidence-based therapeutic strategies for vascular dementia.

## Figures and Tables

**Figure 1 biomedicines-12-01290-f001:**
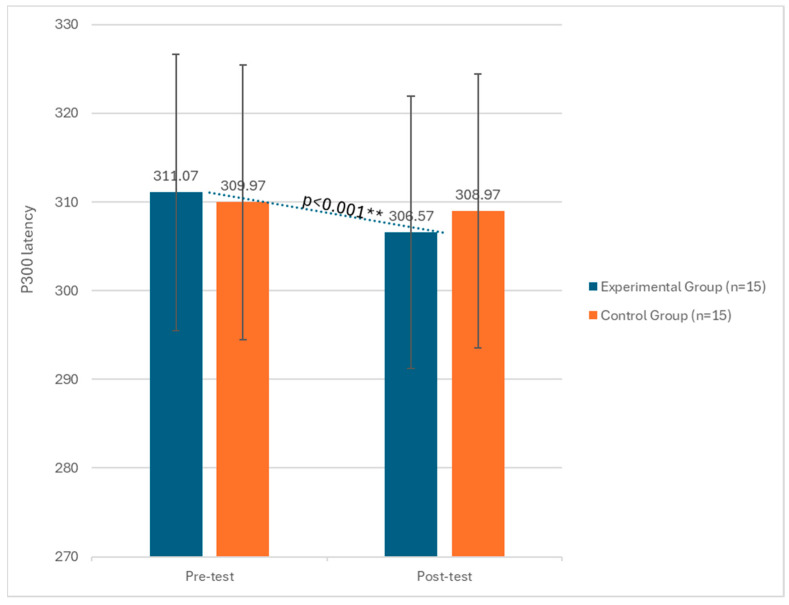
Changes in P300 latency between the pre-test and post-test phases in both the experimental and control groups. ** significant at *p* < 0.001.

**Figure 2 biomedicines-12-01290-f002:**
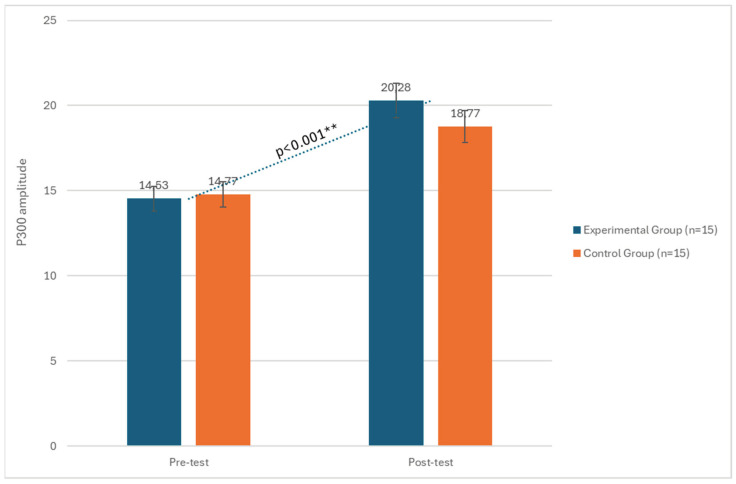
Changes in P300 amplitude between the pre-test and post-test phases in both the experimental and control groups. ** significant at *p* < 0.001.

**Figure 3 biomedicines-12-01290-f003:**
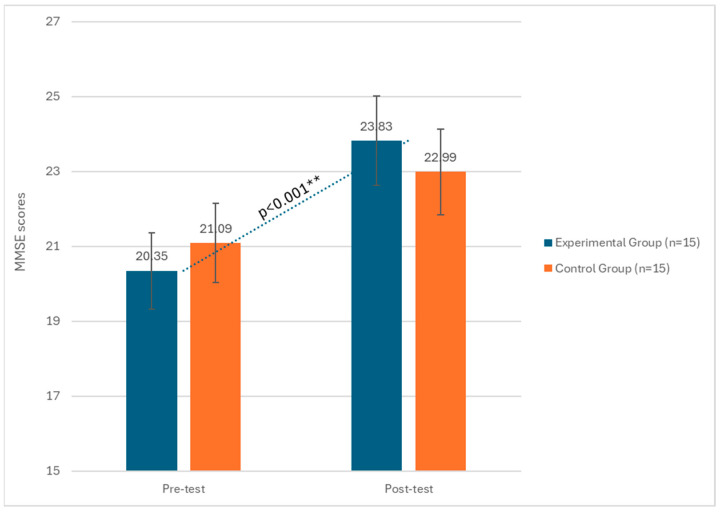
Changes in MMSE scores between the pre-test and post-test phases in both the experimental and control groups. ** significant at *p* < 0.001.

**Table 1 biomedicines-12-01290-t001:** Mean, standard deviation, and t-tests of latency and amplitude of P300 and MMSE.

	Pre-Test	Post-Test			
	M (±SD)	M (±SD)	t	*p*	d
Experimental Group					
MMSE	20.35 (±2.67)	23.83 (±1.88)	7.22	0.001 **	0.78
P300 Latency	311.07 (±2.43)	306.57 (±2.37)	6.81	0.001 **	0.79
P300 Amplitude	14.53 (±2.51)	20.28 (±2.21)	12.05	0.001 **	0.85
Control Group					
MMSE	21.09 (±3.98)	22.99 (±2.56)	1.25	0.23	0.45
P300 Latency	309.87 (±3.11)	308.99 (±3.21)	1.06	0.43	0.39
P300 Amplitude	14.77 (±3.65)	19.77 (±3.18)	0.98	0.45	0.49

** significant at *p* < 0.001.

## Data Availability

Data will be available upon request to the corresponding author, with attention to privacy preservation.

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
