# Peer review of "Does Transcranial Direct Current Stimulation Affect Potential P300-Related Events in Vascular Dementia? Considerations from a Pilot Study"

_biomedicines, 2024, doi:10.3390/biomedicines12061290_

Round 1

Reviewer 1 Report

Comments and Suggestions for Authors

biomedicines-3008420: “Does Transcranial Direct Current Stimulation affect Potential P300-Related Events in Vascular Dementia? Considerations from a pilot study”.

In this study, the authors demonstrate a beneficial potential of the transcranial direct current stimulation (tDCS) in patients with vascular dementia by use of P300 event-related potential (P300 ERP) as a characteristic of their cognitive abilities.

By using of P300 ERP, associated with neurophysiological aspects of learning and memory processes, and tDCS, directly affecting the neuronal activity, is original and promising approach for detailed analysis of basic mechanisms of tDCS effects addressing the gap in the field.

The Mini-Mental State Examination approach used in this study confirms a tight correlation of the beneficial effects of tDCS on cognitive recovery in patients with vascular dementia and P300 results.

The authors concluded that further research needs “larger samples, robust control groups, and long-term follow-up”.

Final conclusions in this study are consistent with the evidence and arguments presented.

Minor remarks:

1)    in Abstract (lines 23,24), abbreviations of “EPR” and “MMSE” should be open;

2)    in line 89, “...], there...”;

3)    in line 147 and below, “MMSE”;

4)    in line 185, “...70dB, SPL),...”;

5)    in lines 184-186, the sentence might be compressed to avoid duplicated parameters;

6)    in line 192, “2.3.3. Mini-Mental State Examination” while “MMSE” in the next line would be better;

7)    in Figures 1 and 2, the plates should be denoted by the letters;

8)    in line 273, the statement needs a reference;

9)    the list of references should be formatted correctly.

Author Response

Dear Editor,

here following we added all the replies point by point.

Reviewer 1

In this study, the authors demonstrate a beneficial potential of the transcranial direct current stimulation (tDCS) in patients with vascular dementia by use of P300 event-related potential (P300 ERP) as a characteristic of their cognitive abilities. By using of P300 ERP, associated with neurophysiological aspects of learning and memory processes, and tDCS, directly affecting the neuronal activity, is original and promising approach for detailed analysis of basic mechanisms of tDCS effects addressing the gap in the field. The Mini-Mental State Examination approach used in this study confirms a tight correlation of the beneficial effects of tDCS on cognitive recovery in patients with vascular dementia and P300 results. The authors concluded that further research needs “larger samples, robust control groups, and long-term follow-up”. Final conclusions in this study are consistent with the evidence and arguments presented.

Reply

Thank you.

Minor remarks:

1)    in Abstract (lines 23,24), abbreviations of “EPR” and “MMSE” should be open;

Reply

Thank you. We modified it.

2)    in line 89, “...], there...”;

Reply

Thank you. We corrected it.

3)    in line 147 and below, “MMSE”;

Reply

Thank you. We modified it.

4)    in line 185, “...70dB, SPL),...”;

Reply

Thank you. We corrected it.

5)    in lines 184-186, the sentence might be compressed to avoid duplicated parameters;

Reply

Thank you. We modified the sentence: 20% of the stimuli consisted of infrequent tones (2 kHz), while 80% comprised frequent tones (1.5 kHz), both with a duration of 200 ms, rise and fall time of 5 ms, and a sound pressure level of 70 dB, SPL (highlighted in yellow).

6)    in line 192, “2.3.3. Mini-Mental State Examination” while “MMSE” in the next line would be better;

Reply

Thank you. We modified it.

7)    in Figures 1 and 2, the plates should be denoted by the letters;

Reply

Thank you. We modified the figures to make them clearer.

8)    in line 273, the statement needs a reference;

Reply

Thank you. We added the reference.

9)    the list of references should be formatted correctly.

Reply

Thank you. We reformatted it.

Reviewer 2

The authors present an interesting study examining the potential of transcranial direct current stimulation as a means of ameliorating the effects of vascular dementia; one of the world’s most prominent forms of dementia that currently has no truly effective cure. Briefly, the authors recruited a number of participants who clinically fit the criteria of having established vascular dementia, and subjected them to daily sessions of the procedure for up to two weeks. Prior to and post treatment, the participants completed a number of validated tests to determine the participants responses, with application of the treatment seemingly demonstrating an improvement. While somewhat limited in its scope, the study highlights the potential of this procedure in alleviating the symptoms of vascular dementia to those afflicted with such, and this possibly warrants further investigation.

In reviewing the manuscript, I made a number of observations. The following should be addressed by the authors when preparing a suitable revision.

  1. The authors note that a ‘control’ group was included but the data displayed in the graphs predominantly focuses on those who received the treatment. While the table does indicate how the control group responded, it would be advised to include the control group in the figures in some way such as to allow comparisons to be drawn between the groups.

Reply

Thank you. We added the control group in the figures.

  1. Can the authors confirm whether there was a statistically significant difference or not in applying the treatment? There is a section which outlines that statistical analysis was performed, but as this is not indicated anywhere with respect to the data, it should be made clearer, even in the text, as to whether the changes noted are significant or not.

Reply

Thank you. We clarified it (highlighted in yellow).

  1. In section 2.4 there is a section labelled intervention – is this the procedure? I would consider moving this as an ‘outcome measure’ and placing it under the ‘procedure’ heading.

Reply

Thank you. We moved the section labelled "intervention" as suggested, placing it under the heading "procedure”.

  1. The n-number should be clearly indicated in each figure/table and/or figure/table legend.  

Reply

Thank you. We modified the figures to make them clearer.

  1. The axis in each figure needs labelling.

Reply

Thank you. We added it.

  1. The writing in the piece is good for the most part, but there are a couple of typos within i.e. the word ‘methods’ in the abstract needs to be italicised to be consistent with the formatting style. The authors should comb the manuscript for instances such as these and address them in advance of any resubmission.

Reply

Thank you. We made the corrections as indicated.

Comments on the Quality of English Language

For the most part the writing is of a very good standard, however, there are a few minor typos within that have been alluded to in the main report. Overall, a good effort, but a quick sweep of the manuscript should address those that are outstanding.

Reply

Thank you. We revised the text as suggested.

Reviewer 2 Report

Comments and Suggestions for Authors

The authors present an interesting study examining the potential of transcranial direct current stimulation as a means of ameliorating the effects of vascular dementia; one of the world’s most prominent forms of dementia that currently has no truly effective cure. Briefly, the authors recruited a number of participants who clinically fit the criteria of having established vascular dementia, and subjected them to daily sessions of the procedure for up to two weeks. Prior to and post treatment, the participants completed a number of validated tests to determine the participants responses, with application of the treatment seemingly demonstrating an improvement. While somewhat limited in its scope, the study highlights the potential of this procedure in alleviating the symptoms of vascular dementia to those afflicted with such, and this possibly warrants further investigation.

In reviewing the manuscript, I made a number of observations. The following should be addressed by the authors when preparing a suitable revision.

1.      The authors note that a ‘control’ group was included but the data displayed in the graphs predominantly focuses on those who received the treatment. While the table does indicate how the control group responded, it would be advised to include the control group in the figures in some way such as to allow comparisons to be drawn between the groups.

2.      Can the authors confirm whether there was a statistically significant difference or not in applying the treatment? There is a section which outlines that statistical analysis was performed, but as this is not indicated anywhere with respect to the data, it should be made clearer, even in the text, as to whether the changes noted are significant or not.

3.      In section 2.4 there is a section labelled intervention – is this the procedure? I would consider moving this as an ‘outcome measure’ and placing it under the ‘procedure’ heading.

4.      The n-number should be clearly indicated in each figure/table and/or figure/table legend.  

5.      The axis in each figure needs labelling.

6.      The writing in the piece is good for the most part, but there are a couple of typos within i.e. the word ‘methods’ in the abstract needs to be italicised to be consistent with the formatting style. The authors should comb the manuscript for instances such as these and address them in advance of any resubmission.

Comments on the Quality of English Language

For the most part the writing is of a very good standard, however, there are a few minor typos within that have been alluded to in the main report. Overall, a good effort, but a quick sweep of the manuscript should address those that are outstanding. 

Author Response

(The authors gave the same response as above.)

Round 2

Reviewer 2 Report

Comments and Suggestions for Authors

The authors have addressed some of my comments, but some measures, while relatively minor and described as being addressed, remain outstanding. 

For example, I requested that the n-number be noted in each figure, and yet this has not been addressed but the authors have noted that they have. Similarly, I asked for labels on the figures - this has not been addressed yet the authors indicate they have, and there is no indication of statistical analyses on the graphs themselves, yet the authors indicate they have addressed this. 

As such, I am retaining the recommendation of major revisions until these have been adequately addressed.  

Author Response

Dear Editor,

here following we added all the replies point by point.

Reviewer 2

The authors have addressed some of my comments, but some measures, while relatively minor and described as being addressed, remain outstanding. 

For example, I requested that the n-number be noted in each figure, and yet this has not been addressed but the authors have noted that they have.

Reply

Thank you. We indicated the n-number in each figure as requested.

Similarly, I asked for labels on the figures - this has not been addressed yet the authors indicate they have, and there is no indication of statistical analyses on the graphs themselves, yet the authors indicate they have addressed this. As such, I am retaining the recommendation of major revisions until these have been adequately addressed.  

Reply

Thank you. We did it. We hope that the work now meets the required standards.

Round 3

Reviewer 2 Report

Comments and Suggestions for Authors

The authors have suitably addressed my comments.